# Unveiling Neural Combinatorial Optimization Model Representations Through Probing

## Abstract

Neural combinatorial optimization (NCO) models have achieved remarkable performance, yet their learned underlying representations remain largely unclear. This hinders real-world application, as industrial stakeholders may want a deeper understanding of NCO models before committing resources. In this paper, we make the first step towards interpreting NCO models by investigating embeddings learned by various architectures through three probing tasks. Specifically, we analyze representative and state-of-the-art attention-based models, including AM, POMO, and LEHD, on the representative Traveling Salesman Problem and Capacitated Vehicle Routing Problem. Our findings reveal that NCO models encode linear representations of Euclidean distances between nodes, while also capturing additional knowledge that help avoid making myopic decisions. Furthermore, we show that architectural choices affect the ability of deep models to accurately represent Euclidean distances and to incorporate non-myopic decision-making strategies. We also verify to what extent NCO models understand the feasibility of constraints. Our work represents an initial effort to interpret NCO models, enhance understanding of why certain architectures outperform others, and demonstrate probing as a valuable tool for analyzing their internal mechanisms.

## 1 Introduction

Recently, learning-based neural combinatorial optimization (NCO) methods have achieved remarkable performance on classic combinatorial optimization problems, such as routing, that is comparable to, or even surpasses, specialized heuristic algorithms designed for these problems (e.g., Concorde (Applegate et al., 2006), LKH3 (Helsgaun, 2017), HGS (Vidal, 2022), etc.). However, the underlying reasons behind these impressive results, particularly the nature of the knowledge learned by these neural models, remain largely unexplored and unclear.

Due to this lack of understanding, current research often relies on final performance metrics, such as average objective function values, to retrospectively assess the strengths and weaknesses of different NCO architectures. This retrospective evaluation approach, however, may lack rigor and precision. Various external influences, such as differing inference strategies (e.g., greedy, sampling, beam search, or specialized methods like Random Re-Construct from Luo et al. (2023)), can lead to significant differences in performance (Zhou et al., 2024). This obscures the assessment of the true representational capacity of NCO models and hinders understanding of how effectively they capture decision-supporting information. Misinterpreting the model architecture, in turn, can negatively affect future model design. Therefore, addressing this gap is crucial.

To address this gap, for the first time, we bring the tool, *probing*, from the computer vision (CV) and natural language processing (NLP) fields to the NCO field, so as to more directly explore the representational capacity of neural network embeddings (Alain & Bengio, 2016; Adi et al., 2016; Belinkov, 2022). Probing involves training auxiliary prediction tasks using the embeddings learned by a pre-trained deep learning model. In the context of NLP, for example, if a simple model, particularly a linear model, can be trained to predict linguistic information about a word (e.g., its part-of-speech tag) or a pair of words (e.g., their semantic relation) from the embeddings, we can reasonably conclude that the embeddings encode this information (for more details, see Liu et al. (2019)).

Unlike NLP tasks, which naturally have intuitive subtasks (e.g., part-of-speech tagging, semantic relation tagging) suitable for probing, combinatorial optimization (CO) problems typically lack such

directly applicable subtasks. To deal with this, we designed probing tasks tailored to evaluate the representational capacity of NCO models in specific CO problems and constructed corresponding datasets. To demonstrate the effectiveness of probing in NCO, we selected two groups of NCO models with similar architectures (both based on the transformer (Vaswani, 2017) structure) but with contrasting design principles. In addition to uncovering certain knowledge embedded in their representations through probing, we also compared the differences in this knowledge captured by the embeddings, which result from the structural variations between the two groups of models. Specifically, one group consists of the classical models AM (Kool et al., 2018) and POMO (Kwon et al., 2020), while the other is their successor LEHD (Luo et al., 2023), which introduces a contrasting architecture—the light encoder heavy decoder (LEHD) model structure, proposed as a potentially more effective alternative to the heavy encoder light decoder (HELD) structure of the earlier models.

Experimental results in Luo et al. (2023) show that the LEHD model indeed outperforms AM and POMO in solving the traveling salesman problem (TSP) and capacitated vehicle routing problem (CVRP), specifically in terms of the objective function value, as measured by the average traveling distances of the routing solutions. Luo et al. (2023) attribute this to the LEHD's ability to better capture the dynamic relationships between nodes of varying sizes. Through our probing tasks, we provide additional and more direct evidence identifying specific factors that might make the LEHD structure superior to the HELD one. These factors include improved perception of Euclidean distances between nodes, a stronger ability to avoid myopic decision-making, and a more robust capability to capture information related to constraints. Unlike final performance metrics, which are influenced by inference strategies, these factors offer strong support for the design idea behind LEHD models. This provides subsequent researchers with a clearer basis for determining whether to incorporate such structural choices in their designs, enhancing confidence in these design decisions.

**Contribution.** Our contributions are as follows: (1) For the first time, we pioneer the use of *probing* in the NCO field to explore and understand the embeddings learned by NCO models. (2) Akin to other impactful probing research in non-NLP fields that lack natural subtasks for probing (Li et al., 2022; Gurnee & Tegmark, 2023), we design targeted probing tasks and create corresponding datasets. (3) We provide evidence that NCO models are capable of capturing knowledge relevant to decision-making in routing problems. (4) By analyzing the differences in the knowledge learned, we shed light on why state-of-the-art models achieve superior performance.

Overall, we take an important first step in unveiling the internal mechanisms of NCO model embeddings through probing techniques. We demonstrate how probing, as a toolkit, can be used to verify why models are effective in CO problems and gain insights into model architecture design. This analysis toolkit supports future work in understanding the representations of black-box NCO models, providing more direct evidence beyond final problem results for performance exploration.

## 2 PROBING TASKS

Since combinatorial optimization problems do not have suitable subtasks to serve as probing tasks, targeted task design is necessary for the specific CO problem being explored. Using the TSP problem as an example, we propose two probing tasks to investigate NCO models: whether the model can perceive the Euclidean distance between nodes (*probing task 1*); and whether the model can learn to avoid constructing solutions in a myopic manner, such as greedily connecting to the nearest node (*probing task 2*). Additionally, we also introduce a probing task using the CVRP problem, examining whether NCO models can capture constraints (*probing task 3*).

**Probing task 1: Euclidean distance**  When solving routing problems in Euclidean space, the Euclidean distance between nodes is a critical piece of information for all solution methods. For instance, a simple greedy algorithm for solving the TSP starts at an arbitrary node, computes the Euclidean distance between the current node and all unvisited nodes, and selects the nearest one as the next destination. This process is repeated until all nodes are visited, returning to the starting node to form a Hamiltonian cycle. In traditional methods, whether using exact approaches (mathematical programming) that rely on the distance matrix of nodes as input or approximate (heuristic) methods (Reinelt, 2003; Liu et al., 2023), the Euclidean distance between any two nodes must be precomputed or computed on the fly. Therefore, for a TSP solver, recognizing the Euclidean distances between nodes is essential. Based on this, we aim to explore whether a trained learning-based NCO model

can capture this critical Euclidean distance between the current node and any of the candidate nodes in its representations.

**Probing task.** *Probing task 1* aims to examine whether the embeddings of NCO models encode the distance between the current node and any of the candidate nodes during decision-making. Given the embeddings of two nodes, a probing model is trained to directly predict the Euclidean distance between them. This probing task, which takes two embeddings as input features, is similar to the probing tasks used in NLP to evaluate pairwise relations between words (Liu et al., 2019).

**Dataset.** Figure 1 illustrates the process of creating a sample for *probing task 1* and its corresponding dataset. Given the current node $n_i$ and any randomly selected node $n_j$ from the candidate nodes, we extract their embeddings $h_i$ and $h_j$ from the relevant layers of the NCO model we want to probe. The embeddings of the two nodes are then concatenated into a feature vector $[h_i, h_j]$, with the Euclidean distance between $n_i$ and $n_j$ serving as the label. By collecting sufficient data in this manner, we construct the dataset for *probing task 1*. Since the label (i.e., the distance) is a continuous, *probing task 1* is framed as a regression prediction task.

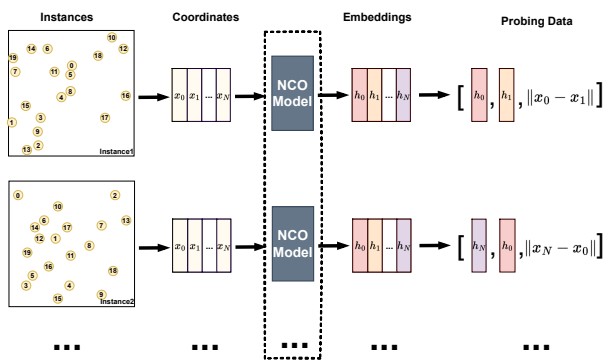

Figure 1: The process of creating the dataset for *probing task 1* is illustrated from left to right. For a given instance, we input its complete data (all nodes) into the NCO model being probed (with the dashed box representing the same NCO model). We then extract the embeddings from the probed part (e.g., the encoder or decoder) or layer of the model and select the corresponding embeddings of the required nodes as features. These features, combined with the label, form a single data point.

**Probing task 2: Avoidance of Myopia** Selecting the next unvisited node solely based on the nearest Euclidean distance, as in the greedy algorithm, will not result in the optimal solution from a global perspective. This approach is often described as "myopic", and many efforts have been made to avoid such shortsighted strategies (Bellman, 1958; Hart et al., 1968; Chekuri & Pal, 2005; Meliou et al., 2007). A well-designed NCO model must similarly learn to avoid myopic strategies and adopt a more global perspective to solve the problem effectively. To investigate this, we design *probing task 2* to explore whether the embeddings of NCO models exhibit the ability to avoid shortsighted decisions at a given step.

**Probing task.** We define *probing task 2* as a binary classification task, where the probing model is trained to determine whether the current node (e.g., $n_i$) should be linked to node $n_j$. Node $n_j$ could either be a myopic choice that leads to a local optimum or the node connected to $n_i$ in the global optimal solution. To assess whether the NCO models make myopic decisions by choosing the nearest Euclidean distance, we construct data points as illustrated in Figure 2.

**Dataset.** First, we randomly generate an instance with $N$ nodes, input it into a mathematical programming model, and use the Gurobi (Gurobi Optimization, LLC, 2024) solver to obtain the theoretical optimal solution, as shown in Figure 2(a). Next, starting from each node, we use a greedy algorithm to generate $N$ solutions and select the best one (as illustrated in Figure 2(b), gradually comparing the next node selected by the greedy algorithm with the optimal solution. For example, in the instance shown in Figure 2, when the current node is node 4, the optimal solution selects node 3, whereas the greedy algorithm selects the nearest one, node 5. Ultimately, we obtain two data points for this instance: node 4 connected to node 3 represents the optimal choice, labeled as a positive example (i.e., the feature is $[h_4, h_3]$ and the label is 1), while node 4 connected to node 5 represents the myopic choice of the greedy algorithm, labeled as a negative example (i.e., the feature is $[h_4, h_5]$ and the label is 0).

**Domain knowledge.** Unlike the relatively straightforward probing tasks and datasets in CV and NLP, probing in the CO field requires incorporating domain-specific knowledge. For instance, in this dataset, there may be multiple optimal solutions. Suppose one of them includes node 4 connected to node 5, which would render a label of 0 incorrect. To verify this, we add a constraint to the mathematical model that forces the connection between nodes 4 and 5. The new optimal solution obtained under this constraint is worse than the original solution without the constraint. Similarly, for data labeled 1, we add a constraint preventing the connection between nodes 4 and 3, and the resulting solution is also worse. This confirms that both labels are valid.

**Probing task 3: Perception of Constraints** For the TSP problem, the first two probing tasks provide a comprehensive analysis of the representational capacity of NCO models. However, for more complex VRP, where additional constraints are introduced, we are curious whether NCO models can capture these constraint-related information. If not, it suggests that NCO models might merely rely on masking to artificially limit their outputs. This would imply an inherent limitation in how NCO models handle constraints.

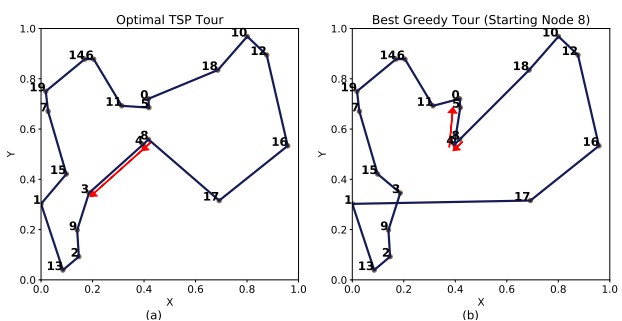

Figure 2: An example of solutions to the TSP for a specific instance: (a) represents the optimal solution generated by the mathematical model and solved using Gurobi; (b) shows the best solution obtained through a greedy algorithm.

**Probing task.** To answer this question, we design *probing task 3* to explore whether NCO models can capture the knowledge required to determine the feasibility of the capacity constraint in the CVRP problem. Since the capacity constraint primarily involves the linear (additive) relationship among the demands of nodes, we design *probing task 3* to check whether the embeddings of two nodes can represent the sum of their demands. Thus, for *probing task 3*, a probing model is trained to predict the sum of the demands given the embeddings of two nodes.

**Dataset.** We extract the embeddings of two nodes, $h_i$ and $h_j$, from the relevant layers of the NCO model being probed. Unlike the previous non-linear probing tasks, predicting the sum of two demands—a linear addition task—may be inherently too simple. Therefore, a linear probing model might not be sufficient to demonstrate whether the NCO model can capture this knowledge. To delve deeper, in addition to concatenating the embeddings of the two nodes ($[h_i, h_j]$) as the input for the probing task, we also apply Hadamard product on the two embeddings, $[h_i \odot h_j]$, as an alternative input. The latter approach aims to simulate the attention computation process in attention-based NCO models (as in most models where the decoder ultimately uses attention to compute a compatibility score to determine node selection probability), allowing us to examine whether the model can capture the additive effect of demand features.

## 3 MODELS

**Probing Model** For all three probing tasks, we use a linear model for the corresponding regression and classification tasks. Specifically, we train a simple linear fully connected (FC) layer for both classification and regression tasks. If this linear model can accurately predict the probing tasks based on the embeddings from the NCO models, it indicates that the knowledge relevant to the probing tasks can be easily extracted from the embeddings (Alain & Bengio, 2016; Liu et al., 2019). This also suggests that the NCO model, from which the embeddings for the probing tasks are derived, has the ability to encode this knowledge in its representations after training.

It is important to note that a linear model cannot directly capture the nonlinear relationship of Euclidean distance. For *probing task 1* (the Euclidean distance regression task), a linear model would have no explanatory power if the input only consists of the nodes' coordinate information. In the

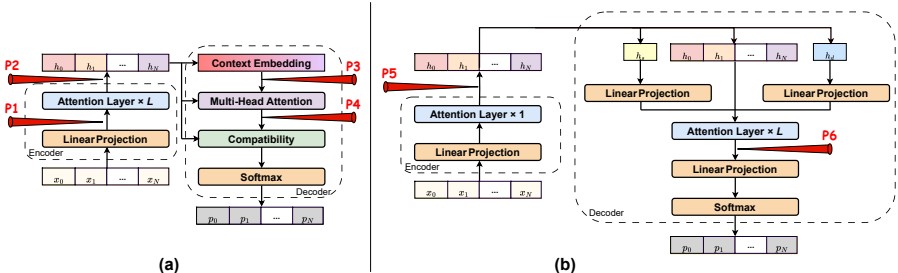

Figure 3: The figure illustrates the architecture of two NCO models: (a) represents the HELD structure, as seen in AM and POMO, while (b) represents the LEHD structure. The red arrows in the figure indicate the positions where we probe the model, extracting the embeddings.

most extreme case, where the input for *probing task 1* (i.e., the features) are solely the two nodes' coordinates, the regression model's $R^2$ value would be zero, because the covariance between the label and the linear model's output is zero. This result is also reflected in the experimental findings presented later in Section 5.1. However, when the probing model's $R^2$ value is greater than 0, and the closer it is to 1, the stronger the evidence that the NCO model has the ability to perceive Euclidean distances. This indicates that the information related to Euclidean distance, encoded in the model's embeddings, can be linearly extracted, thereby validating the NCO model's ability to effectively represent this relationship.

**NCO Models**   We selected three NCO models for probing: AM (Kool et al., 2018), POMO (Kwon et al., 2020), and LEHD (Luo et al., 2023). Figure 3 (adapted from the original figures in their respective papers) illustrates the architecture of these models. Detailed descriptions of these models can be found in the Appendix B. Through the three probing tasks described above, we demonstrate that these models are capable of representing decision-related knowledge relevant to routing problems. Additionally, by comparing the differences in their embeddings across the three probing tasks and analyzing the architectural differences between the models, we explore the reasons why the higher-performing models exhibit superior results in terms of the final objective function value.

## 4   EXPERIMENTAL SETUP

**Datasets**   In line with the problem settings of AM, POMO, and LEHD, we select 20 nodes as a small-scale instance and 100 nodes as a relatively large-scale instance. This setup allows us to use pre-trained models and extract embeddings from models corresponding to these scales. We will make all the code and datasets publicly available for future research.

**Routing instances.** For *probing task 1* and *probing task 2*, we generate 10,000 TSP instances with 20 nodes and 10,000 instances with 100 nodes, respectively, following the method introduced in AM (Kool et al., 2018), which was subsequently used by both POMO and LEHD. Since *probing task 2* requires global optimal solutions and greedy solutions to create the probing dataset labels, we use Gurobi to solve the optimal solution for these 20,000 instances. For the greedy solution, we perform a greedy algorithm starting from each node. Then, we generate the datasets for these 20,000 instances following the method described in Section 2.

For *probing task 3*, we similarly generate 10,000 instances with 20 nodes and 10,000 instances with 100 nodes following the method used in AM. The aim of this paper is to pioneer the application of probing to the study of NCO models. Therefore, as an initial exploration, we have conducted only one constraint-related probing task on the CVRP. We believe that future probing research on CVRP can offer deeper insights into how NCO models handle constraints. If further labeling of CVRP solutions is required, we recommend using the HGS (Vidal, 2022) to solve CVRP instances.

**Probing datasets.** After generating the routing problem instances, we input them into the NCO model to extract embeddings. For a finer-grained analysis, we extract embeddings from different layers and positions, as indicated by the red arrows in Figure 3. We provide a detailed explanation

Table 1: Comparison of probing task results for NCO models. The underlined results indicate they are derived from the final node embeddings of the three models, with further details available in Figure 8 in the Appendix C.

| | | Probing input | Probing task 1 | | | Probing task 2 | | | | | Probing task 3 | | |
|---|---|---|---|---|---|---|---|---|---|---|---|---|---|
| | | | RMSE | MAE | $R^2$ score | Accuracy | Precision | Recall | F1 score | AUC | RMSE | MAE | $R^2$ score |
| 20 node embeddings | w/o ints. | AM-Init | 0.2452 | 0.2028 | -0.0003 | 49.28% | 0.49 | 0.47 | 0.48 | 0.49 | 0.0000 | 0.0000 | 1.0000 |
| | | AM-Enc-$l1$ | 0.2066 | 0.1665 | 0.2899 | 66.90% | 0.70 | 0.59 | 0.64 | 0.72 | 0.0088 | 0.0070 | 0.9945 |
| | | AM-Enc-$l3$ | 0.2119 | 0.1711 | 0.2529 | 70.43% | 0.73 | 0.65 | 0.69 | 0.76 | 0.0273 | 0.0219 | 0.9471 |
| | | AM-Enc-$l3$-w/c | 0.2140 | 0.1724 | 0.2381 | 69.30% | 0.72 | 0.63 | 0.67 | 0.75 | - | - | - |
| | | AM-Enc-$l3$-w/g | 0.2134 | 0.1721 | 0.2423 | 70.97% | 0.74 | 0.65 | 0.69 | 0.76 | - | - | - |
| | | POMO-Enc-$l1$ | 0.2115 | 0.1711 | 0.2558 | 64.50% | 0.68 | 0.56 | 0.61 | 0.67 | - | - | - |
| | | POMO-Enc-$l6$ | 0.2196 | 0.1787 | 0.1981 | 70.10% | 0.71 | 0.67 | 0.69 | 0.76 | - | - | - |
| | | LEHD-Enc-$l1$ | 0.2115 | 0.1719 | 0.2554 | 64.08% | 0.68 | 0.53 | 0.60 | 0.67 | 0.0038 | 0.0030 | 0.9990 |
| | | LEHD-Dec-$l1$ | 0.0062 | 0.0046 | 0.9994 | 74.10% | 0.79 | 0.66 | 0.72 | 0.78 | 0.0100 | 0.0078 | 0.9929 |
| | | LEHD-Dec-$l6$ | 0.0590 | 0.0451 | 0.9421 | 78.25% | 0.79 | 0.77 | 0.78 | 0.86 | 0.0366 | 0.0288 | 0.9047 |
| | w/ ints. | AM-Init | 0.1318 | 0.1000 | 0.7111 | 51.95% | 0.52 | 0.47 | 0.50 | 0.52 | 0.0000 | 0.0000 | 1.0000 |
| | | AM-Enc-$l1$ | 0.0235 | 0.0171 | 0.9908 | 70.28% | 0.74 | 0.63 | 0.68 | 0.77 | 0.0269 | 0.0209 | 0.9488 |
| | | AM-Enc-$l3$ | 0.0657 | 0.0514 | 0.9282 | 75.90% | 0.78 | 0.73 | 0.75 | 0.83 | 0.0955 | 0.0767 | 0.3533 |
| | | AM-Enc-$l3$-w/c | 0.0653 | 0.0512 | 0.9291 | 74.95% | 0.77 | 0.72 | 0.74 | 0.82 | - | - | - |
| | | AM-Enc-$l3$-w/g | 0.0660 | 0.0518 | 0.9275 | 75.67% | 0.78 | 0.72 | 0.75 | 0.83 | - | - | - |
| | | POMO-Enc-$l1$ | 0.0543 | 0.0430 | 0.9510 | 69.23% | 0.72 | 0.62 | 0.67 | 0.74 | - | - | - |
| | | POMO-Enc-$l6$ | 0.1119 | 0.0890 | 0.7917 | 78.88% | 0.79 | 0.80 | 0.79 | 0.86 | - | - | - |
| | | LEHD-Enc-$l1$ | 0.0424 | 0.0325 | 0.9701 | 66.88% | 0.71 | 0.57 | 0.63 | 0.72 | 0.0112 | 0.0087 | 0.9910 |
| | | LEHD-Dec-$l1$ | 0.0069 | 0.0052 | 0.9992 | 74.12% | 0.79 | 0.67 | 0.72 | 0.79 | 0.0159 | 0.0125 | 0.9820 |
| | | LEHD-Dec-$l6$ | 0.0592 | 0.0452 | 0.9417 | 78.55% | 0.80 | 0.77 | 0.78 | 0.86 | 0.0632 | 0.0502 | 0.7169 |
| 100 node embeddings | w/o ints. | AM-Init | 0.2498 | 0.2084 | -0.0012 | 50.48% | 0.51 | 0.45 | 0.48 | 0.50 | 0.0000 | 0.0000 | 1.0000 |
| | | AM-Enc-$l1$ | 0.2186 | 0.1791 | 0.2332 | 56.00% | 0.57 | 0.53 | 0.55 | 0.60 | 0.0137 | 0.0110 | 0.9878 |
| | | AM-Enc-$l3$ | 0.2212 | 0.1800 | 0.2151 | 66.30% | 0.68 | 0.61 | 0.65 | 0.71 | 0.0237 | 0.0187 | 0.9635 |
| | | AM-Enc-$l3$-w/c | 0.2245 | 0.1830 | 0.1915 | 67.10% | 0.69 | 0.62 | 0.65 | 0.72 | - | - | - |
| | | AM-Enc-$l3$-w/g | 0.2224 | 0.1806 | 0.2062 | 65.88% | 0.68 | 0.60 | 0.64 | 0.71 | - | - | - |
| | | POMO-Enc-$l1$ | 0.2210 | 0.1799 | 0.2166 | 57.60% | 0.59 | 0.53 | 0.55 | 0.62 | 0.0199 | 0.0157 | 0.9743 |
| | | POMO-Enc-$l6$ | 0.2231 | 0.1825 | 0.2014 | 71.83% | 0.72 | 0.72 | 0.72 | 0.79 | 0.0445 | 0.0356 | 0.8710 |
| | | LEHD-Enc-$l1$ | 0.2194 | 0.1796 | 0.2280 | 55.93% | 0.56 | 0.54 | 0.55 | 0.60 | 0.0052 | 0.0040 | 0.9950 |
| | | LEHD-Dec-$l1$ | 0.0094 | 0.0068 | 0.9986 | 67.45% | 0.72 | 0.57 | 0.64 | 0.72 | 0.0069 | 0.0055 | 0.9913 |
| | | LEHD-Dec-$l6$ | 0.0469 | 0.0370 | 0.9647 | 76.50% | 0.77 | 0.75 | 0.76 | 0.85 | 0.0178 | 0.0140 | 0.9426 |
| | w/ ints. | AM-Init | 0.1334 | 0.1033 | 0.7143 | 51.82% | 0.52 | 0.47 | 0.49 | 0.53 | 0.0000 | 0.0000 | 1.0000 |
| | | AM-Enc-$l1$ | 0.0262 | 0.0193 | 0.9890 | 63.80% | 0.66 | 0.57 | 0.61 | 0.68 | 0.0356 | 0.0278 | 0.9177 |
| | | AM-Enc-$l3$ | 0.0444 | 0.0339 | 0.9684 | 69.08% | 0.72 | 0.63 | 0.67 | 0.76 | 0.0645 | 0.0510 | 0.7290 |
| | | AM-Enc-$l3$-w/c | 0.0587 | 0.0463 | 0.9448 | 70.33% | 0.73 | 0.66 | 0.69 | 0.77 | - | - | - |
| | | AM-Enc-$l3$-w/g | 0.0447 | 0.0340 | 0.9679 | 69.15% | 0.72 | 0.63 | 0.67 | 0.75 | - | - | - |
| | | POMO-Enc-$l1$ | 0.0276 | 0.0212 | 0.9877 | 66.15% | 0.68 | 0.60 | 0.64 | 0.71 | 0.0272 | 0.0214 | 0.9517 |
| | | POMO-Enc-$l6$ | 0.0802 | 0.0640 | 0.8968 | 72.47% | 0.72 | 0.73 | 0.73 | 0.80 | 0.1157 | 0.0951 | 0.1281 |
| | | LEHD-Enc-$l1$ | 0.0421 | 0.0325 | 0.9716 | 61.82% | 0.63 | 0.59 | 0.61 | 0.66 | 0.0069 | 0.0053 | 0.9915 |
| | | LEHD-Dec-$l1$ | 0.0075 | 0.0054 | 0.9991 | 67.20% | 0.72 | 0.57 | 0.63 | 0.73 | 0.0086 | 0.0068 | 0.9867 |
| | | LEHD-Dec-$l6$ | 0.0468 | 0.0367 | 0.9648 | 77.00% | 0.78 | 0.76 | 0.77 | 0.85 | 0.0308 | 0.0243 | 0.8280 |

of these extracted embeddings in Section C.2. Each probing dataset is split into training and test sets, with all reported results based on the test set, i.e., out-of-sample data.

**Evaluation metrics** To evaluate the performance of the probes, we utilize a variety of standard metrics. For the regression tasks, we report metrics such as root mean square error (RMSE), mean absolute error (MAE), and the coefficient of determination ($R^2$). For the classification tasks, the evaluation metrics include accuracy, precision, recall, F1 score, and the area under the curve (AUC).

# 5 EMPIRICAL RESULTS

Table 1 presents the results of the three probing tasks. We introduce the "Probing input" column in Appendix C.2, which represents the embeddings extracted from different positions in the NCO models. Figure 9 compares results across different layers, while Figure 10 highlights the changes during training. More detailed results are provided in Appendix C.

## 5.1 RESULTS AND DISCUSSION FOR PROBING TASK 1

For *Probing Task 1*, we examine the ability of the three NCO models to linearly represent the Euclidean distances between pairs of nodes (specifically, the current node and any unvisited node) during decision-making, by training linear probes and evaluating their performance. First, we demonstrate that all three NCO models can linearly capture the Euclidean distances between nodes. Then, through comparison, we find that LEHD performs better in accurately perceiving Euclidean distances than POMO and AM, particularly in the robustness of its representation extraction method, as the inclusion or exclusion of interaction terms has little impact on LEHD's probing performance.

Based on this observation, we analyze the possible reasons and provide additional experimental results, leading to a key insight for NCO model design. Finally, by comparing the ability of different layers to represent Euclidean distances, we validate why it is necessary to explicitly introduce Euclidean distance information into NCO models.

**Existence.** As shown in the "w/o ints." rows of AM-Init in Table 1 for both 20-node and 100-node examples, the values indicate that the initial embeddings of AM fail to capture the nonlinear relationship of Euclidean distance (with $R^2$ values close to 0). These embeddings are derived by mapping the raw features, specifically the 2D coordinates in the TSP, through a linear projection into the shared dimensional space of the encoder and decoder (128 dimensions for all three models discussed in this paper). In Section 3, we explained that the $R^2$ of a Euclidean distance regression model using node coordinates as input is zero because it cannot capture the nonlinear nature of Euclidean distance. As a result, the initial embeddings essentially retain the properties of the raw features and similarly fail to linearly capture Euclidean distances. The phenomenon of an $R^2$ value of zero for the initial embeddings can be observed across all NCO models.

However, after passing through the NCO model, the $R^2$ values for AM, POMO, and LEHD increase to 0.2529, 0.1981, and 0.9421, respectively, for 20-nodes example. Even when considering interaction terms, the $R^2$ values for all three models' embeddings after the encoder or decoder are significantly higher than those of the initial embeddings, approaching 1 in 100-node example. This indicates that the representations in these NCO models contain linearly decodable Euclidean distance information, meaning they have learned how to linearly represent Euclidean distances.

**Comparison of HELD and LEHD.** In the results shown in Table 1, for both the 20-node and 100-node instances, LEHD's approach—using a single encoder layer followed by multi-layer attention calculations between the current node and other nodes in the decoder—outperforms the embedding method used by AM and POMO, where all nodes are embedded through multiple encoder layers, in perceiving Euclidean distances. Notably, without interaction terms, AM and POMO both struggle to accurately perceive Euclidean distances. Additionally, as shown in the results for "AM-Enc-$l3$-w/c" and "AM-Enc-$l3$-w/g," even with the extra information provided by context embeddings or glimpse embeddings, AM and POMO do not improve the accuracy of perceiving the Euclidean distance between the current node and other nodes.

LEHD's recalculation of the embeddings of candidate nodes in its decoder, through the attention mechanism with the current node embedding, may allow it to more effectively capture the relationships between the current node and other nodes. Specifically, as shown in the decoder of Figure 3(b), the embedding of the current node, $h_s$, participates in the attention calculations with the remaining nodes after passing through a linear projection, updating their embeddings. In contrast to AM and POMO, which treat all node embeddings equally and perform node embedding only once, LEHD's decoder design allows for a more accurate perception of the distances between the current node and the remaining nodes. To verify this, we conducted additional experiments on LEHD, and the results are presented in Table 2.

First, we extract the embeddings of two remaining nodes for probing and find that the probe achieves an $R^2$ of only 0.0927. This indicates that LEHD is indeed more focused on the relationship between the current node and other nodes. Additionally, when we probe the embedding from the linear projection below $h_s$ in the decoder (Figure 3(b), before the attention calculation), its $R^2$ dropped to 0.2555, significantly lower than the original 0.9421. This suggests that the attention mechanism in LEHD's decoder is crucial for accurately capturing the Euclidean distances between the current node and the other nodes.

Table 2: Supplementary experiments for LEHD. The first two rows show distance perception between non-current nodes and others, while the last row shows the effect of removing attention from LEHD.

| Probing input | RMSE | MAE | $R^2$ score |
|---|---|---|---|
| LEHD-Dec-$l1$-other | 0.2091 | 0.1694 | 0.2620 |
| LEHD-Dec-$l6$-other | 0.2318 | 0.1898 | 0.0927 |
| LEHD-Dec-w/o-att | 0.2115 | 0.1719 | 0.2555 |

This leads to an insight for future NCO model: recalculating node embeddings through the attention mechanism in the decoder enables more accurate perception of Euclidean distances than relying solely on context embeddings, as in the case of AM and POMO, to provide current information.

**As Layers Deepen.** By comparing the results (including both node-scale instances and whether interaction terms are used) of the same model across different layers, we find that the ability of the embeddings to perceive Euclidean distances decreases as the number of attention layers increases in all three models. Notably, after six attention layers, POMO shows a more significant decline in Euclidean distance perception compared to AM, which has the same structure but only three attention layers. This suggests that while deeper attention layers may enhance other decision-making capabilities (as discussed in the next section), the model's ability to perceive distances diminishes.

In subsequent research based on AM/POMO models, some models introduce node distance information to enhance performance: either by explicitly incorporating distance information to adjust the model's output (Wang et al., 2024), or by designing distance-aware attention mechanisms (Zhou et al., 2024). Through probing experiments, we verify that these approaches introduce Euclidean distance to mitigate its perception deficiency as the number of layers increases in NCO models. This provides important guidance for future improvements to AM and POMO-based models.

## 5.2 RESULTS AND DISCUSSION FOR PROBING TASK 2

Through *Probing Task 2*, we explore whether NCO models can learn to avoid making decisions based solely on distance. First, we demonstrate that the embeddings of all three NCO models exhibit the ability to avoid myopic decision-making. Then, by comparison, we find that LEHD performs better in this regard than POMO, with POMO outperforming AM. This result aligns with their performance on the objective function values in the routing problem and further supports the conclusion from *Probing Task 1* regarding the impact of model structure on performance.

**Existence.** We use the "AM-Init" results as a baseline reference, with AUC values consistently at 0.5, indicating that the initial embeddings cannot linearly extract the knowledge needed to distinguish which nodes are connected to the current ones in the global optimal solutions (namely, the optimal edges). To confirm that *Probing Task 2* is not relying on Euclidean distances for node differentiation, we further examine the initial embeddings with interaction terms, whose AUC values remain close to 0.5, suggesting they still fail to distinguish between global optimal or greedy edges. In contrast, in *Probing Task 1*, the initial embeddings with interaction terms achieve an $R^2$ above 0.7, indicating that the initial embeddings with interaction terms have linear explanatory power for Euclidean distances. This observation confirms that the two probing tasks are fundamentally different. It also implies that if the embeddings in an NCO model can be linearly distinguished in *Probing Task 2*, the model has learned to avoid myopic decision-making and capture the knowledge needed to find the global optimal solution.

**Comparison.** The results in Table 1 show that all three NCO models possess the ability to avoid myopic decision-making, and this ability improves as the number of attention layers increases. Additionally, for the 20-node instance, LEHD's performance in this regard matches POMO (both with an AUC of 0.86) and slightly outperforms AM (0.83). In the 100-node instance, however, LEHD outperforms the others with an AUC of 0.85, compared to 0.80 for POMO and 0.76 for AM. These results are fully consistent with their performance on the optimization problem outcomes. More detailed results can be found in Figure 8 in Appendix C.3.

Furthermore, by observing the results of "AM-Enc-$l$3-w/c" and "AM-Enc-$l$3-w/g", which show almost no difference from "AM-Enc-$l$3" and remain lower than LEHD, we further support the conclusion from *Probing Task 1*: LEHD's heavy decoder structure captures more relevant decision-supporting information compared to AM and POMO, which rely on context embeddings.

## 5.3 RESULTS AND DISCUSSION FOR PROBING TASK 3

Through *Probing Task 3*, using the capacity constraint in the CVRP problem as an example, we demonstrate that probing can be applied to study the ability of NCO models to represent constraints. We observe that, while all three NCO models can capture the linear (additive) relationship between node demands, this ability weakens with an increasing number of layers, similar to the perception of Euclidean distances. This observation is particularly noteworthy in the Hadamard product probing input, $[h_i \odot h_j]$. As discussed in Section 2, we simulate attention calculations using this Hadamard product input. Many NCO models, including AM and POMO, calculate a compatibility score by attention calculations before applying the output Softmax. In this context, the $R^2$ values for the final

output layer decrease significantly compared to the first layer, as shown in the "w/ ints." rows in Table 1. In some results, $R^2$ even drops to the 0.1-0.35 range, indicating that these NCO models may no longer accurately capture whether the demand exceeds vehicle capacity and are likely relying on masking to impose final output modifications and constraints.

This probing task raises an interesting research question for designing NCO models to handle constraints: should additional constraint-related information be incorporated into NCO models, similar to how distance information is added in studies (Wang et al., 2024; Zhou et al., 2024)? Future work can further explore this by designing and implementing more probing tasks to deepen the understanding of how NCO models handle constraints.

## 5.4 PROBING MODEL RESULTS

In addition to the quantitative results mentioned above, we pioneeringly introduce an analysis of the probing model's outcomes. Unlike prior applications of probing in CV, NLP, or other fields, which primarily focused on the performance of the probing model, we delve deeper into the CO domain by analyzing the probing model's coefficients. This analysis reveals differences in how various NCO models' embeddings capture information. Consequently, it helps uncover the reasons why the better-performing NCO models achieve their superior results.

Figure 4 presents the coefficients of the probing models obtained on the test datasets for the two TSP-related probing tasks across the three NCO models. In each subplot, the horizontal axis represents the feature indices, while the vertical axis shows the coefficient values. The upper part of each subplot displays the actual values of the coefficients for the corresponding features. The two colors on the left and right sides represent the embeddings of two nodes: the first 128 dimensions correspond to the embeddings of the first node (the current node), and the latter 128 dimensions correspond to the embeddings of the second node (one of the candidate nodes). The lower part of each subplot illustrates the statistical significance of the respective features.

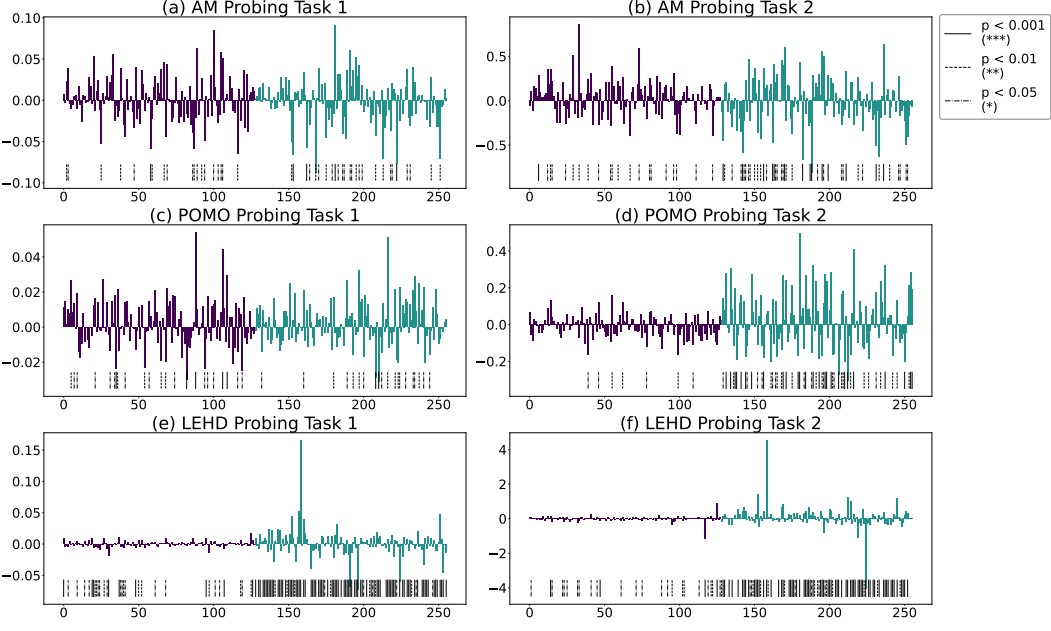

Figure 4: The figure illustrates the coefficients of probing models for two TSP-related probing tasks across all NCO models.

From the results shown in Figure 4, we observe that LEHD, the best-performing model, exhibits more statistically significant features in its node embeddings for both TSP-related probing tasks compared to AM and POMO. Specifically, examining the coefficients of each node's embeddings reveals that for the current node, the probing model's coefficients tend to have smaller absolute

values, with only a subset being statistically significant. In contrast, the embeddings of other nodes (those relevant to the decision-making process for selecting the next node to visit in the current step) have a greater number of statistically significant dimensions. This pattern is also observed in AM and POMO during the myopia-avoidance task, albeit with far fewer statistically significant features in their embeddings compared to LEHD. However, when it comes to perceiving Euclidean distances, the embeddings of AM and POMO as features show no such distinction. In subplots (a) and (c), the coefficients and the number of significant features for the two nodes' embeddings are similar, regardless of their roles as the current node or other nodes.

## 5.5 ROBUSTNESS CHECK

To further validate the robustness of probing as a tool for analyzing NCO models and the probing tasks we designed, we demonstrate *Probing Task 1* for distance perception in non-Euclidean space. Specifically, we selected MatNet (Kwon et al., 2021), a state-of-the-art model designed for solving asymmetric TSP (ATSP).

We use MatNet's row embeddings and column embeddings for pairs of nodes as features, and the distances between the corresponding nodes in the distance matrix as labels to construct the probing dataset. For example, the row embeddings of node $i$ and node $j$ are used as features, with the corresponding label being the value in the distance matrix at the intersection of row $i$ and column $j$, denoted as $dist(i, j)$. Similarly, the column embeddings of node $i$ and node $j$ are used as features, with the label being $dist(j, i)$.

The results are shown in Figure 11. As the number of layers in MatNet increases, the ability of its embeddings to perceive distances improves, with the $R^2$ rising from less than 0.2 in the first layer to approximately 0.5 in the final layer. Additionally, we conduct supplementary comparison experiments. In the first experiment, serving as a baseline, the embeddings of node $i$ and node $j$ are used as inputs, but the labels are replaced with random distance values unrelated to both nodes from the distance matrix. The resulting probing $R^2$ is -0.0232, indicating that the probe could not learn any distance information from random labels based on the embeddings. In the second experiment, we swap the labels between row and column embeddings, assigning the row embeddings of node $i$ and $j$ with the label $dist(j, i)$ and vice versa. The resulting probing $R^2$ is 0.2532. Comparing these results, we conclude the following: MatNet's dual-attention structure effectively learns information from the asymmetric distance matrix. Furthermore, regardless of whether the embeddings of two nodes are correctly aligned, they can still partially represent distance information. However, the model's ability to capture correct distance information between two nodes is significantly stronger than its ability to capture incorrect distance information, with $R^2$ values of approximately 0.5 versus 0.2, respectively.

## 6 CONCLUSION

Using the probing method, we are the first to reveal the representational capabilities of NCO models, thereby deepening the understanding of the internal mechanisms of these black-box models. Through three probing tasks, we find that both classical and state-of-the-art attention-based NCO models can perceive Euclidean distances and have learned to avoid making myopic decisions based solely on distance. Although the ability to perceive Euclidean distances decreases as the number of attention layers increases, the models simultaneously acquire more knowledge to avoid shortsighted decision-making. Similarly, the ability of these NCO models to perceive linear constraints weakens with deeper layers. This finding offers insights and supports for future works focused on improving NCO models' ability to handle constraints. Additionally, by comparing the performance of different models, we uncover that the best-performing SOTA models possess a stronger representational capacity for capturing decision-relevant knowledge.

We believe that with the design and introduction of more probing tasks in the future, the reliability and interpretability of NCO models will further improve, increasing their potential for real-world applications. Additionally, our exploration of combining combinatorial optimization knowledge with probing to understand model representations in the NCO domain provides a method to enhance the potential of deep learning applications in other areas. This approach is expected to promote the broader application of deep learning models in scientific and engineering fields.

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

## A    RELATED WORK

Neural (deep learning-based) methods have been applied to various combinatorial optimization problems for several years (Khalil et al., 2017; Bengio et al., 2021). With the rapid advancement of deep learning (DL), an increasing number of approaches have been introduced to address these classical problems in operations research (OR). In the context of the routing problem discussed in this paper, researchers have explored methods such as graph convolutional networks (GCNs) (Kool et al., 2022; Zhou et al., 2023), pointer networks with recurrent neural networks (RNNs) (Bello et al., 2016; Nazari et al., 2018), diffusion-based approaches (Sun & Yang, 2023), and attention mechanisms (Kool et al., 2018; Lu et al., 2019; Kwon et al., 2020; Ma et al., 2021; Luo et al., 2023), which are the primary focus of this study.

Although NCO methods have seen rapid development in academia, industries remain cautious about deploying them to replace classical OR methods. This is because these DL-based methods are perceived as black-box models, lacking the reliability and interpretability of traditional OR approaches. As a result, even though some NCO models have achieved strong performance on certain instances, they are still met with skepticism. For example, Santana et al. (2023) raises concerns about the overuse of GNNs, noting that the improvements achieved by GNN-based methods over traditional distance-related approaches were minimal. To address this, we are the first to unveil the inner workings of NCO models, aiming to enhance understanding of their internal mechanisms.

The probing method used in this paper was initially applied to understand the representations of DL models in computer vision (Alain & Bengio, 2016) and natural language processing (Adi et al., 2016). Beyond traditional DL tasks, probing has also demonstrated effectiveness in other domains, such as exploring world representations (Li et al., 2022; Gurnee & Tegmark, 2023) in large language models, auditory representations (Shah et al., 2021; Ngo & Kim, 2024), and studying the quality of unsupervised reinforcement learning representations (Zhang et al., 2022).

## B    AM, POMO, LEHD

AM is one of the earliest and most successful learning-based models for routing tasks. It is pioneering in introducing the widely popular Transformer architecture to combinatorial optimization problems, inspiring a multitude of subsequent models. POMO, as a notable example, retains a structure fundamentally similar to AM (with minor differences, such as in context embedding) but introduces a novel reinforcement learning (RL) training method.

AM not only introduces the Transformer architecture but also makes significant contributions to the model design for routing problems. A notable idea is AM's context embedding in the decoder, which focuses on the current node and the starting node (for TSP problems). Although many later models do not adopt this exact context embedding design, the core idea of focusing on these two key nodes remains. For example, even though LEHD's decoder design differs from AM's, it fundamentally considers how to represent information from these two critical nodes.

Specifically, the difference between LEHD and AM lies in their architectural design. Figure 3 illustrates the architecture of both models. In Figure 3(a), AM uses a multi-layer encoder to learn how to represent node information based on their input features (coordinates), while the decoder performs a single attention computation on the node representations generated by the encoder, producing a global "glimpse" for decision-making without updating the node embeddings. This is known as the "Heavy Encoder Light Decoder" structure. In contrast, LEHD adopts a "Light Encoder Heavy Decoder" structure, where the encoder uses only a single attention layer to learn node representations, while the decoder, at each step, re-learns the embeddings of the current node, destination node, and candidate nodes through multiple attention layers. In LEHD, as shown in Figure 3(b), $h_s$ and $h_d$ represent the embeddings of the current node (referred to as the starting node in LEHD) and the destination node, while the node embeddings located in the middle are filtered in LEHD to exclude previously visited nodes.

## C  Detailed results from three probing tasks

### C.1  Analysis of Input Data

For each probing task, we begin by analyzing the input probing dataset, using *Probing Task 1* as an example for dataset exploration and preprocessing. As this is a regression problem, we first analyze the target variable to observe whether the label distribution is skewed, whether there are outliers, and other characteristics. Figure 5 shows the label distribution for the 20-node dataset in *Probing Task 1* and *Probing Task 3*, with the dataset generation process detailed in Section 2. As seen, the distribution of distances between randomly selected nodes after a current node is chosen approximates a normal distribution. The distribution of demand follows a similar pattern. For *Probing Task 2*, we generate one data point with a label of 1 and one with a label of 0 for each routing instance, resulting in a 1:1 label distribution.

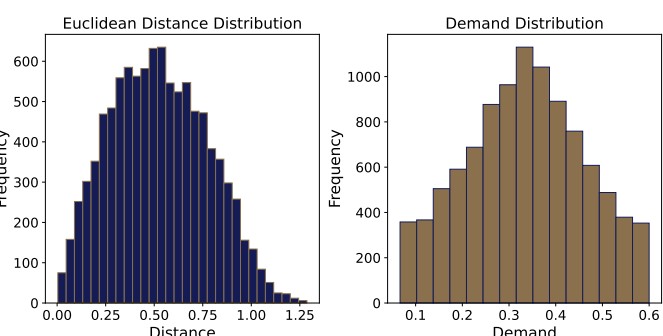

Figure 5: The figure shows the label distribution for a probing datasets in *Probing Task 1* and *Probing Task 3*.

Next, we conducted a feature correlation analysis on the probing dataset. For the probing dataset formed by the embeddings of two nodes (128 dimensions each), there are a total of 256 features. By examining the correlation heatmap in Figure 6, we observe that there are some positive and negative correlations among the 128 dimensions within a single node's embedding, but overall, there are not many strong correlations apart from initial embedding. We can also observe that there are a few scattered stronger correlations in LEHD's embedding, which could be the source of its enhanced ability to retain the perception of Euclidean distance. Additionally, there is no significant correlation between the embeddings of the two nodes.

### C.2  Probing inputs

In the names under the "Probing input" column of Table 1, the first segment (AM, POMO, LEHD) indicates from which NCO model the embeddings are extracted. The second segment (Init., Enc., Dec.) represents the different parts of the NCO model from which the embeddings are extracted: initial embeddings, encoder embeddings, and decoder embeddings, respectively.

In the encoder of the NCO model, the initial embeddings (Init.) are extracted before the attention layer, as shown at position P1 in Figure 3. P2 and P5 represent the embeddings from the encoder's attention layers (Enc.), while P6 represents those from the decoder's attention layers (Dec.). We use "$l$" followed by a number to indicate from which specific layer the embeddings are extracted. Specifically, AM's encoder has three layers, POMO's encoder has six layers, and LEHD's encoder has only one layer, while its decoder has six layers.

Since AM and POMO do not update node embeddings in the decoder, their node embeddings in decoder are not included as probing inputs. However, they introduce context embeddings in the decoder to represent the information needed for routing decisions. For example, in solving TSP, the context embeddings are formed by concatenating the embedding of the starting node $h_s$, the current node embedding, and the graph embedding $h_{graph}$—calculated as the mean of all node embeddings. To explore the representational capacity of this design, we also use the context embeddings $[h_i, h_j, h_s, h_{graph}]$ as probing inputs, denoted as "AM-Enc-$l$3-w/c". Additionally, AM uses the context

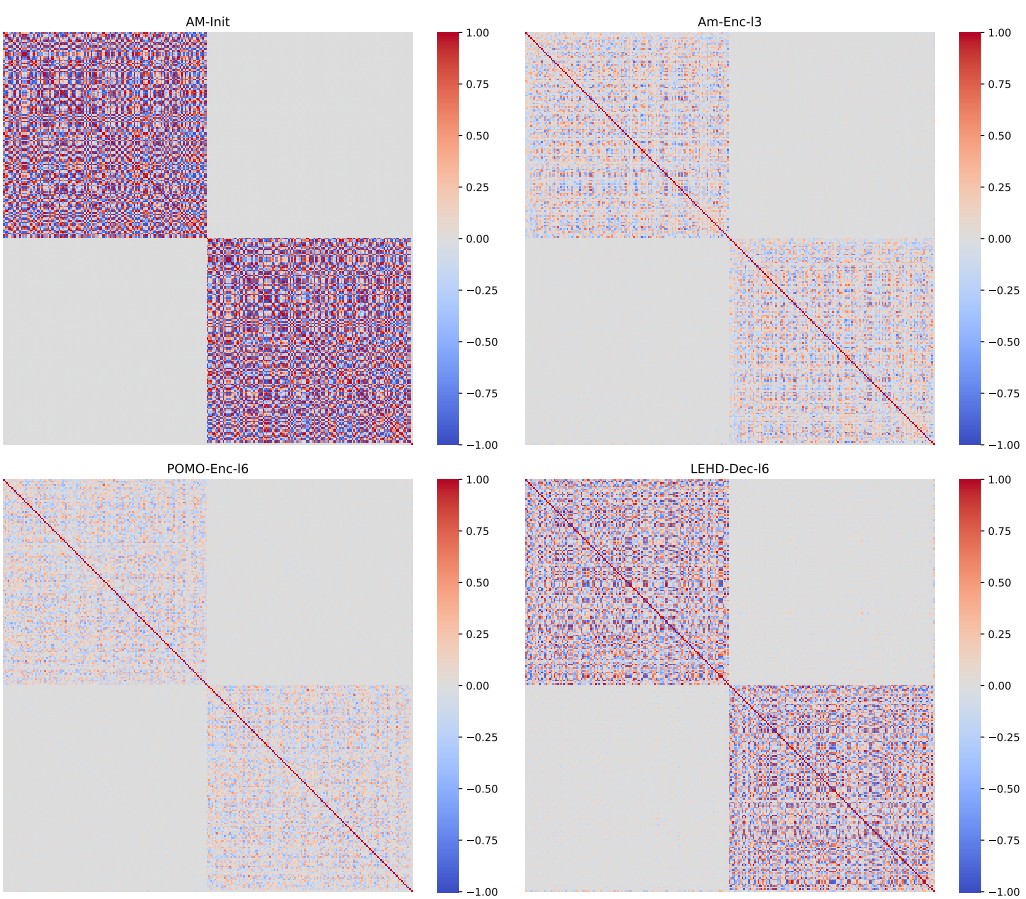

Figure 6: The figure shows the correlation heatmap for all 256 features (comprising two 128-dimensional node embeddings) of the AM encoder embedding, POMO encoder embedding, and the LEHD decoder embedding.

embeddings as a query to compute attention with other node embeddings, generating a glimpse embedding $h_{glimpse}$. To test this, we probe the input $[h_i, h_j, h_{glimpse}]$ and denote it as "AM-Enc-l3-w/g". In Figure 3, P3 and P4 represent the positions where the context embeddings and glimpse embeddings are extracted, respectively.

Additionally, for the first two probing tasks, besides using $[h_i, h_j]$ as input, we also consider the element-wise product of the two node embeddings as an interaction term (Liu et al., 2019), i.e., $[h_i, h_j, h_i \odot h_j]$ as input. Some parts of certain models may linearly combine node embeddings (for instance, many NCO models concatenate the embeddings of nodes and then pass them through a linear projection). In such components of the models, the embeddings are expected to capture decision-relevant information through simple linear combinations. However, embeddings from certain parts in attention-based models, such as those used to compute a compatibility score among node embeddings through attention mechanisms, may behave differently. In this case, relying solely on the linear input $[h_i, h_j]$ may not fully assess the model's representational capacity. Therefore, we introduce the interaction term $h_i \odot h_j$ to emulate the attention computation. We conduct probing experiments with both input methods: "w/o ints." refers to input without interaction terms $[h_i, h_j]$, and "w/ ints." refers to input with interaction terms $[h_i, h_j, h_i \odot h_j]$, as shown in Table 1.

For *Probing Task 3*, we use both $[h_i, h_j]$ and $[h_i \odot h_j]$ as inputs (the rationale is discussed in the *probing task 3* paragraph in Section 2). In Table 1, the "w/o ints." rows correspond to the results for $[h_i, h_j]$, while the "w/ ints." rows correspond to the results for $[h_i \odot h_j]$. Finally, for the 20-node and 100-node instances, we conducted the three probing tasks using NCO models trained on the

corresponding scales. The results for both are grouped and presented in Table 1 and discussed in the following sections.

Table 3, Table 4, and Table 5 present the specific features, labels, and the number of observations for the different inputs across the three probing tasks.

Table 3: The details of Probing inputs of *Probing task 1*.

| | | Probing input | # Observations | Features | Label |
|---|---|---|---|---|---|
| 20 and 100 | w/o ints. | Coordinates | | $[x_i, x_j]$ | |
| | | AM-Init | | $[h_i, h_j]$ | |
| | | AM-Enc-$l1$ | | $[h_i, h_j]$ | |
| | | AM-Enc-$l3$ | | $[h_i, h_j]$ | |
| | | AM-Enc-$l3$-w/c | 10000 | $[h_i, h_j, h_{graph}]$ | |
| | | AM-Enc-$l3$-w/g | | $[h_i, h_j, h_{glimpse}]$ | $\|x_i - x_j\|$ |
| | | POMO-Enc-$l1$ | | $[h_i, h_j]$ | |
| | | POMO-Enc-$l6$ | | $[h_i, h_j]$ | |
| | | LEHD-Enc-$l1$ | | $[h_i, h_j]$ | |
| | | LEHD-Dec-$l1$ | | $[h_i, h_j]$ | |
| | | LEHD-Dec-$l6$ | | $[h_i, h_j]$ | |
| | w/ ints. | Coordinates | | $[x_i, x_j, x_i \odot x_j]$ | |
| | | AM-Init | | $[h_i, h_j, h_i \odot h_j]$ | |
| | | AM-Enc-$l1$ | | $[h_i, h_j, h_i \odot h_j]$ | |
| | | AM-Enc-$l3$ | | $[h_i, h_j, h_i \odot h_j]$ | |
| | | AM-Enc-$l3$-w/c | | $[h_i, h_j, h_{graph}, h_i \odot h_j]$ | |
| | | AM-Enc-$l3$-w/g | 10000 | $[h_i, h_j, h_{glimpse}, h_i \odot h_j]$ | $\|x_i - x_j\|$ |
| | | POMO-Enc-$l1$ | | $[h_i, h_j, h_i \odot h_j]$ | |
| | | POMO-Enc-$l6$ | | $[h_i, h_j, h_i \odot h_j]$ | |
| | | LEHD-Enc-$l1$ | | $[h_i, h_j, h_i \odot h_j]$ | |
| | | LEHD-Dec-$l1$ | | $[h_i, h_j, h_i \odot h_j]$ | |
| | | LEHD-Dec-$l6$ | | $[h_i, h_j, h_i \odot h_j]$ | |

Table 4: The details of Probing inputs of *Probing task 2*.

| | | Probing input | # Observations | Features | Label |
|---|---|---|---|---|---|
| 20 and 100 | w/o ints. | AM-Init | | $[h_i, h_j]$ | |
| | | AM-Enc-$l1$ | | $[h_i, h_j]$ | |
| | | AM-Enc-$l3$ | | $[h_i, h_j]$ | |
| | | AM-Enc-$l3$-w/c | | $[h_i, h_j, h_{graph}]$ | |
| | | AM-Enc-$l3$-w/g | 20000 | $[h_i, h_j, h_{glimpse}]$ | Binary |
| | | POMO-Enc-$l1$ | | $[h_i, h_j]$ | |
| | | POMO-Enc-$l6$ | | $[h_i, h_j]$ | |
| | | LEHD-Enc-$l1$ | | $[h_i, h_j]$ | |
| | | LEHD-Dec-$l1$ | | $[h_i, h_j]$ | |
| | | LEHD-Dec-$l6$ | | $[h_i, h_j]$ | |
| | w/ ints. | AM-Init | | $[h_i, h_j, h_i \odot h_j]$ | |
| | | AM-Enc-$l1$ | | $[h_i, h_j, h_i \odot h_j]$ | |
| | | AM-Enc-$l3$ | | $[h_i, h_j, h_i \odot h_j]$ | |
| | | AM-Enc-$l3$-w/c | | $[h_i, h_j, h_{graph}, h_i \odot h_j]$ | |
| | | AM-Enc-$l3$-w/g | 20000 | $[h_i, h_j, h_{glimpse}, h_i \odot h_j]$ | Binary |
| | | POMO-Enc-$l1$ | | $[h_i, h_j, h_i \odot h_j]$ | |
| | | POMO-Enc-$l6$ | | $[h_i, h_j, h_i \odot h_j]$ | |
| | | LEHD-Enc-$l1$ | | $[h_i, h_j, h_i \odot h_j]$ | |
| | | LEHD-Dec-$l1$ | | $[h_i, h_j, h_i \odot h_j]$ | |
| | | LEHD-Dec-$l6$ | | $[h_i, h_j, h_i \odot h_j]$ | |

Table 5: The details of Probing inputs of *Probing task 3*. $d_i$ denotes the demand for node $i$.

| | | Probing input | # Observations | Features | Label |
|---|---|---|---|---|---|
| 20 and 100 | w/o ints. | AM-Init | | $[h_i, h_j]$ | |
| | | AM-Enc-$l$1 | | $[h_i, h_j]$ | |
| | | AM-Enc-$l$3 | | $[h_i, h_j]$ | |
| | | POMO-Enc-$l$1 | | $[h_i, h_j]$ | |
| | | POMO-Enc-$l$6 | 10000 | $[h_i, h_j]$ | $d_i + d_j$ |
| | | LEHD-Enc-$l$1 | | $[h_i, h_j]$ | |
| | | LEHD-Dec-$l$1 | | $[h_i, h_j]$ | |
| | | LEHD-Dec-$l$6 | | $[h_i, h_j]$ | |
| | w/ ints. | AM-Init | | $[h_i \odot h_j]$ | |
| | | AM-Enc-$l$1 | | $[h_i \odot h_j]$ | |
| | | AM-Enc-$l$3 | | $[h_i \odot h_j]$ | |
| | | POMO-Enc-$l$1 | | $[h_i \odot h_j]$ | |
| | | POMO-Enc-$l$6 | 10000 | $[h_i \odot h_j]$ | $d_i + d_j$ |
| | | LEHD-Enc-$l$1 | | $[h_i \odot h_j]$ | |
| | | LEHD-Dec-$l$1 | | $[h_i \odot h_j]$ | |
| | | LEHD-Dec-$l$6 | | $[h_i \odot h_j]$ | |

### C.3 PLOTS OF THE PROBING TASK 2 RESULTS

For *Probing Task 2*, we use AM as an example to plot the training loss when using the initial embeddings and the encoder (layer 3) embeddings as probing inputs. As shown in Figure 7, the loss for the encoder embeddings converges to a lower value. The final evaluation results, e.g., AUC, also indicate that the encoder embeddings perform better in classification compared to the initial embeddings.

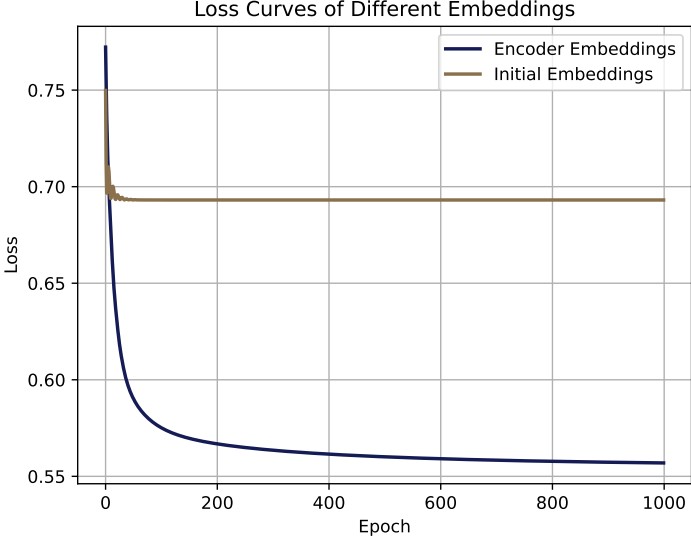

Figure 7: The figure shows the training loss for the initial embedding and encoder embedding of the AM model.

Figure 8 provides more detailed results for *Probing Task 2*. The third row shows the results for LEHD, where it consistently achieves better classification performance on both the 20-node and 100-node instances, regardless of whether interaction terms are used. Additionally, by analyzing the classification of positive and negative examples, we can further understand the differences in how

embeddings capture both the nearest distance and the global optimal solution. A detailed analysis of positive and negative cases reveals the extent to which the model mistakenly identifies the nearest node as the optimal solution node, leading to shortsighted (myopic) decisions.

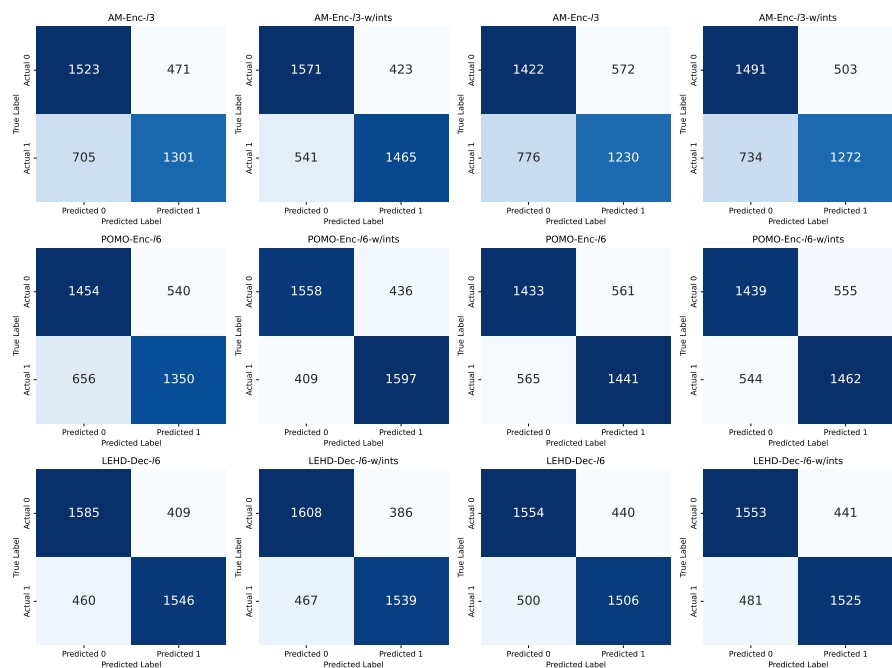

Figure 8: The confusion matrices for *Probing Task 2* across the three models. The left two columns represent the results for instances with 20 nodes, while the right two columns correspond to instances with 100 nodes.

## C.4 PLOTS FOR DETAILED PROBING RESULTS

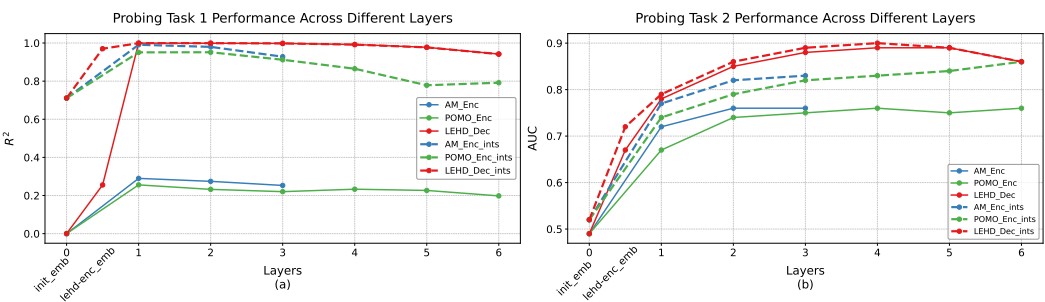

Figure 9: The figure illustrates the probing results across different layers.

In Figure 9, we present the results of two TSP probing tasks across different layers of embeddings for three trained models. As shown in Figure 9(a), the initial embeddings (obtained by linearly projecting the coordinates into a high-dimensional space) of all three models exhibit weak Euclidean distance perception. However, after passing through just one attention layer, all models achieve highly accurate Euclidean distance perception. This ability slightly diminishes as model depth increases.

Despite this slight decline in Euclidean distance perception, NCO models learn additional capabilities that enable better decision-making. For instance, the ability to avoid myopic node selection

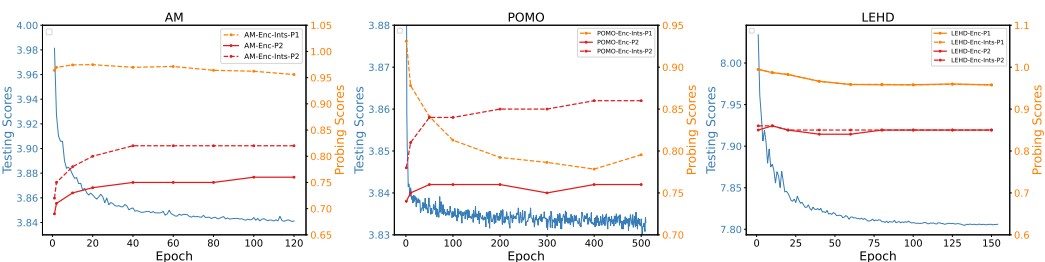

Figure 10: The figure illustrates the probing results during NCO models training.

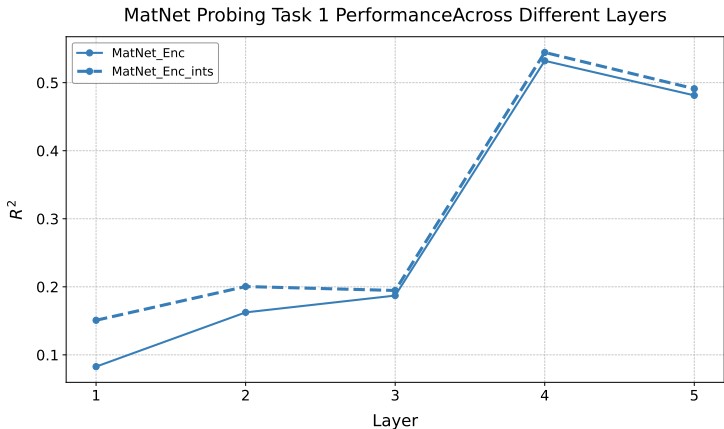

Figure 11: The figure illustrates the probing results of MatNet across different layers.

improves with increased model depth, as illustrated in Figure 9(b). An exception is observed in the last two layers of LEHD, where the ability to avoid myopic decisions slightly decreases, potentially indicating that the model has learned more complex decision-making strategies. Future research could further explore this phenomenon and what LEHD learns in its deeper layers. Overall, through these two probing tasks, we demonstrate that when NCO models solve TSP problems, they can perceive Euclidean distances (low-level features) in shallow layers and learn a decision space beyond the Euclidean distance space (high-level features) in deeper layers. In this decision space, NCO models can avoid making myopic decisions.

Figure 10 illustrates the evolution of results for two TSP probing tasks during the training process of the three NCO models. As shown, for AM and POMO, the model performance improvement during the initial epochs is the fastest, and the results for *Probing Task 2* (related to avoiding myopic decisions) also improve rapidly in early learning epochs. This alignment suggests a positive correlation between *Probing Task 2* results and the models' final performance. In contrast, LEHD achieves peak performance on *Probing Task 2* right from the start of training, indicating that the LEHD model has already learned how to avoid myopic decisions early in the process. What additional information LEHD learns to make its node selection decisions could be further investigated in future research by designing new probing tasks.

The probing results of MatNet are shown in Figure 11, with a detailed analysis provided in Section 5.5.

## C.5 MORE POMO RESULTS

Table 6 presents additional experiments on POMO. The first row corresponds to the original POMO data from Table 1, the second row represents the embeddings of a POMO model trained using supervised learning (SL), and the third row corresponds to POMO augmented with context embeddings.

Table 6: More POMO Probing results.

| | | Probing input | Probing task 1 | | | Probing task 2 | | | | |
|---|---|---|---|---|---|---|---|---|---|---|
| | | | RMSE | MAE | $R^2$ score | Accuracy | Precision | Recall | F1 score | AUC |
| 20 nodes | w/o | POMO-Enc-$l6$ | 0.2196 | 0.1787 | 0.1981 | 70.10% | 0.71 | 0.67 | 0.69 | 0.76 |
| | | POMO-SL-Enc-$l6$ | 0.2183 | 0.1770 | 0.2073 | 70.50% | 0.72 | 0.66 | 0.69 | 0.76 |
| | | POMO-Enc-$l6$-w/c | 0.2250 | 0.1809 | 0.1876 | 69.75% | 0.71 | 0.67 | 0.69 | 0.76 |
| | w | POMO-Enc-$l6$ | 0.1119 | 0.0890 | 0.7917 | 78.88% | 0.79 | 0.80 | 0.79 | 0.86 |
| | | POMO-SL-Enc-$l6$ | 0.1044 | 0.0825 | 0.8186 | 76.35% | 0.78 | 0.74 | 0.76 | 0.84 |
| | | POMO-Enc-$l6$-w/c | 0.1189 | 0.0942 | 0.7732 | 78.97% | 0.79 | 0.80 | 0.79 | 0.86 |
| 100 nodes | w/o | POMO-Enc-$l6$ | 0.2231 | 0.1825 | 0.2014 | 71.83% | 0.72 | 0.72 | 0.72 | 0.79 |
| | | POMO-SL-Enc-$l6$ | 0.2219 | 0.1809 | 0.2102 | 72.65% | 0.73 | 0.72 | 0.72 | 0.81 |
| | | POMO-Enc-$l6$-w/c | 0.2249 | 0.1818 | 0.1646 | 71.25% | 0.72 | 0.72 | 0.72 | 0.79 |
| | w | POMO-Enc-$l6$ | 0.0802 | 0.0640 | 0.8968 | 72.47% | 0.72 | 0.73 | 0.73 | 0.80 |
| | | POMO-SL-Enc-$l6$ | 0.0797 | 0.0638 | 0.8980 | 73.60% | 0.74 | 0.73 | 0.74 | 0.81 |
| | | POMO-Enc-$l6$-w/c | 0.0807 | 0.0645 | 0.8923 | 72.28% | 0.72 | 0.73 | 0.72 | 0.80 |

The results show that the SL-trained POMO achieves similar probing task results to the RL-trained POMO. This observation aligns with the findings from the ablation study in Luo et al. (2023), where SL-trained and RL-trained POMO models exhibit comparable performance.

## C.6 JSSP PRECEDENCE CONSTRAINT

In addition to the routing problem analyzed earlier, we also apply probing to test the precedence constraints in the Job-shop Scheduling Problem (JSSP). For JSSP, we evaluate a classic learning model Zhang et al. (2020), which is based on a graph neural network. The datasets for this probing task are constructed as follows: we extract embeddings for all nodes, pair two node embeddings that satisfy the precedence constraint with a label of 1 ($[h_i,h_j]$-1), and pair two node embeddings that violate the constraint with a label of 0 ($[h_m,h_n]$-0). As an ablation, we also construct an alternative dataset where pairs that satisfy the precedence constraint are incorrectly labeled as 0: $[h_i,h_j]$-1, $[h_n,h_m]$-0.

The results show that for the correct dataset, the probing model achieves an AUC of 1, while for the ablation dataset, the AUC is 0.5. This indicates that the NCO model effectively captures the precedence constraint information between nodes in its embeddings. Here, we provide an initial demonstration of how probing can explore the NCO model's perception of constraints in the JSSP. In the future, more sophisticated probing tasks can be designed to further analyze how the NCO model perceives constraints and incorporates them into its decision-making process, thereby offering deeper insights into the design of NCO models.

