# OpenReview forum: "Unveiling Neural Combinatorial Optimization Model Representations Through Probing"
_ICLR.cc/2025/Conference — Submitted to ICLR 2025_

### Official Review · Reviewer_sr65 · 2024-11-02

**Soundness:** 2
**Presentation:** 3
**Contribution:** 2
**Rating:** 5
**Confidence:** 4

**Summary:**

The paper introduces the use of a well-known technique, probing, in the analysis of NCO models. Specifically, it introduces three probing tasks for Euclidean routing CO problems and applies them to analyze the embeddings of three NCO models applied to solving TSP and CVRP.

**Strengths:**

- Probing is novel in the NCO domain and could potentially help understand the internal mechanisms of NCO models.
- The idea of performing probing at different positions within the network seems very useful, it can help to understand how embeddings are transformed across layers.
- The paper is very well-written and easy to understand.

**Weaknesses:**

- Chosen architectures can handle only Euclidean TSP and CVRP. As a result, the probing tasks are too simple. The most interesting question is how NCO models represent and handle constraints, but the only constraint in this study is the simple capacity constraint of CVRP in probing task 3. Even with this simple constraint, there is no clear answer - the conclusion is that NCO models "likely" rely on masking. It would be valuable to apply the same approach to more complex CO problems with additional constraints, such as CVRP with time windows or JSSP (to probe precedence constraints).

- The answer to probing task 2 "whether the NCO models exhibit the ability to avoid shortsighted decisions at a given step" is trivial. All NCO models perform better than a greedy search, so it is clear that they **have** the ability to avoid myopic decisions. However, the main question is **how** they do that, but this study does not provide an answer. This still remains inside a black box. I think these simple probing tasks are not sufficient to provide an answer; deeper analysis is required. This could involve analyzing attention patterns or investigating specific steps where the models significantly deviate from greedy choices.

- A simple linear connected layer may not be powerful enough to extract relevant knowledge from embeddings. If the fully connected layer does not accurately predict a probing task, it does not necessarily mean that there is no relevant knowledge — it could mean that it is difficult to detect the relationships between embeddings, and a more powerful network may find them.

- Experiments show that there is no correlation between probing tasks and the final performance of the model.

**Questions:**

Related to the concerns mentioned above, I have additional questions/remarks:

1. I suggest trying a more powerful network for the prediction of probing tasks and comparing the results with those obtained from simple linear models. This would certainly eliminate any doubt about the inefficiency of the FC network.

2. The variety of architectures should be enriched. POMO and AM, in fact, share the same architecture, while LEHD uses a very similar attention-based architecture with a difference in the decoder. This work would be much more impactful if it also applied probing techniques to some of the recent transformer-based models designed to handle edge features within the model, like [1], [2], or some similar approach for non-Eucledian routing problems. It will be valuable to how these models preserve (or forget) distance information, which is provided explicitly. Are there any limitations or challenges to applying your method to more complex architectures?

3. Experiments show that POMO performs worse than AM in capturing the capacity constraint (at the end of the encoder, R² values are 0.87/0.13 vs. 0.96/0.72, respectively). However, when we compare the overall results of POMO and AM (in terms of optgap), POMO performs twice as well as AM. Given this, what is the background for the claim in line 493 that "we should integrate the constraint-related information into deeper layers"? Will it really help in better decisions, or maybe degrade performance? Can we conclude from the experiments for POMO/AM/LEHD that "forgetting capacity constraints" is useful for decision-making or not?

4. The motivation for using an interaction term in the experiments remains unclear to me. While it is true that "some NCO models concatenate embeddings..." none of the models in this study do so.

5. Why are some values missing from Table 1 / Probing task 3? Esp. POMO-Enc-l1,6 for 20 nodes are missing, while the same values for 100 nodes exist.

[1] Kwon et al. Matrix Encoding Networks for Neural Combinatorial Optimization, NeurIPS 2021

[2] Drakulic et al. GOAL: A Generalist Combinatorial Optimization Agent Learning, arXiv:2406.15079

---

> ### Comment · Reviewer_sr65 · 2024-11-26
>
> Thank you very much for your responses.
>
> In particular, I appreciate the additional experiment with MatNet. However, I am not sure why you chose a non-Euclidean space with an asymmetric distance matrix. MatNet also works for Euclidean TSP as demonstrated in their Table D.1. and that would be more appropriate for comparing results with your existing models. But, it is only a detail.
>
> I was particularly curious to see how models that explicitly incorporate distance matrices perform on Probing Task 1, and I can say the results align with expectations - MatNet explicitly "injects" distance information at every layer, and its ability to perceive distances improves in higher layers. However, MatNet remains inferior to AM+POMO on the TSP, as shown in Table D.1 and this again raises the question of the relation between probing task 1 and the model's performance.
>
> I believe that probing can be useful for demystifying the black box of NCO, but current tasks are too simple to provide some answers. A broader range of problems and models needs to be covered, along with the development of more sophisticated probing tasks.
>
> After reviewing all the rebuttal responses and updated manuscript, I will increase my score to 5.

---

> > ### Author Response · Authors · 2024-11-27
> >
> > We sincerely appreciate your thoughtful recognition of the adjustments we’ve made at this stage. Your feedback is invaluable to us, and it means a great deal to see our efforts acknowledged. Additionally, we are more than willing to engage in further discussions to address the concerns you currently have regarding the current version of this paper.
> >
> > 1.The rationale behind using MatNet for ATSP is as follows:
> >
> > * Firstly, compared to TSP, MatNet was specifically designed for ATSP. We also wanted to explore whether probing could be applied to non-Euclidean spaces, which is why we titled this section in the revised manuscript as Robustness Check.
> > * Secondly, as the MatNet authors stated, TSP as a special case of ATSP for MatNet, where the distance matrix is symmetric—essentially, the upper and lower triangular parts of the matrix are identical. We think that MatNet's distance perception ability might be independent of whether the distance matrix is symmetric. To test this, we just conducted an additional experiment: we used the MatNet pretrained on ATSP (since the code and pretrained model for the Appendix D experiments mentioned in the original MatNet GitHub repository were not provided) to solve TSP and extract embeddings. For Probing Task 1, which examines the model's distance perception ability, we found that the results were quite similar to those on ATSP. The final layer achieved a result of **0.5345**, which is lower than the results previously obtained with POMO under the same experimental setup (0.7917).  This aligns with the performance shown in Table D1 of MatNet's original paper, where MatNet's performance on Euclidean distance TSP was inferior to that of POMO. Therefore, we believe this outcome highlights the correlation between the model's final performance and the probing task. (Again, there are numerous factors influencing the final performance of the model, and Probing Task 1 explores only one of them.)
> >
> > 2.Regarding the concern that the current three probing tasks might be simple, we would like to share our further thoughts:
> >
> > * **Foundation for Probing Analysis of NCO Models**. We believe that these three probing tasks can serve as a foundational tasks to using probing for analyzing NCO models. For example, the first two probing tasks for TSP effectively demonstrate that NCO models can perceive distances with reasonable accuracy in their final embeddings (prior to the softmax layer that outputs probabilities). They also show, to varying degrees, the model's ability to avoid falling into myopic decision-making based solely on distances. We consider these two abilities to be particularly important for solving TSP problems.
> > * **Initial Motivation Behind Our Work**. The primary goal of our paper is to introduce probing as a tool for analyzing models in the NCO domain using these three intuitive and easily understandable probing tasks. We believe that this example will inspire future work to design more sophisticated probing tasks for exploring various aspects of NCO models. Over time, such efforts can enhance the interpretability of NCO models or provide valuable insights to improve their overall performance.
> >
> > We once again sincerely thank you for your valuable feedback and the opportunity to further explain our approach. If there are any remaining concerns or further points you would like to discuss, please do not hesitate to let us know. We look forward to hearing your thoughts!

---

### Official Review · Reviewer_C5Uc · 2024-11-04

**Soundness:** 2
**Presentation:** 2
**Contribution:** 2
**Rating:** 5
**Confidence:** 4

**Summary:**

This study explores the interpretability of neural combinatorial optimization (NCO) models by analyzing the embeddings learned by various architectures through three probing tasks. It reveals that the NCO models encode Euclidean distances between nodes while also capturing additional knowledge to avoid myopic decisions. Furthermore, this research demonstrates that probing is a valuable tool for analyzing the internal mechanisms of these models.

**Strengths:**

The paper effectively incorporates domain knowledge from the field of combinatorial optimization to construct probing tasks that assist in the interpretability analysis of NCO models.
2.  The paper successfully applies probing techniques from the NLP field to the study of NCO model interpretability.

**Weaknesses:**

Major Weaknesses:
1.	The manuscript lacks sufficient details about the applied probing models, making the experimental results unconvincing. For instance, in line 236, the authors indicate that they trained a simple linear fully connected network as a probing model. However, they do not provide information on the construction of the training set and specific training procedures. Additionally, there is no clarification on how the probing models accommodate inputs of differing dimensionalities.
2.	The experimental content is inadequate to support the authors' conclusions:
a)  The claim that LEHD outperforms HELD due to "LEHD’s recalculation of the embeddings" (line 407) is not substantiated. This is because Table 1 only includes the AM-Enc-w/c and AM-Enc-w/g cases and does not provide results for POMO-Enc-w/c and POMO-Enc-w/g.
b)  Regarding the impact of increasing model layers on performance, the analysis of the HELD models are limited to layers l1 and l3, neglecting results for layer l2.
c)  The conclusion that recalculation enhances the performance of the NCO model (line 409) is solely based on experiments with the LEHD model. However, there are no corresponding results provided for the HELD model.
Minor Weaknesses:
1.	In constructing the dataset for probing task 2, the authors introduce constraints to avoid multiple optimal solutions. However, in line 175, they state that "the new optimal solution obtained under this constraint is worse than the original solution without the constraint," which undermines the dataset's credibility.
2.	In the analysis of the TSP task, the authors conclude that "the ability of the embeddings to perceive Euclidean distances decreases as the number of attention layers increases in all three models" (line 432). Conversely, they also state that “deeper layers enhance the embeddings' ability to avoid myopic decision-making” (line 466).  This presents a contradiction regarding the effects of attention layer depth on embedding performance.
3.	Many evaluation metrics used in the experimental section, such as MSE, RMSE, and R², are unnecessary as their positive or negative correlations are evident.
4.	One of the primary models analyzed, AM, is cited from arXiv and dates back to 2018, making it relatively outdated (line 577).

**Questions:**

1.	How do you train and use a probing model on the regression and classification tasks?
2.	What’s the performance of POMO-Enc-w/c and POMO-Enc-w/g? Moreover, please compare it with the LEHD models.
3.	What’s the performance of AM-Enc-l2 and POMO-Enc-(l2-l5)?
4.	Can recalculating node embeddings methods enhance the performance of HELD models, i.e., AM and POMO?
5.	How does the domain constraints to probing task 2 affect the quality of constructed dataset?
6.	For the TSP task, what layer model should we use: a deeper one or a shallower one?

---

### Official Review · Reviewer_B7Vv · 2024-11-04

**Soundness:** 2
**Presentation:** 2
**Contribution:** 3
**Rating:** 6
**Confidence:** 3

**Summary:**

This paper provides analysis results of state-of-the-art attention-based neural combinatorial optimization (NCO) models. Specifically, this paper investigates embeddings learned by various architectures by proposing tree probing tasks. Probing task 1 examines whether the embeddings of NCO models can encode the Euclidean distance between nodes. Probing task 2 checks if the embeddings of NCO models can enable the ability to avoid myopic decisions to fine a global optimal solution. Probing task 3 tests whether the embeddings of NCO models can capture constraints in CO problems. For each probing task, this paper constructs a dataset, and trains a probing model on each dataset. This paper mainly analyzes two NCO architectures: (1) AM and POMO, and (2) LEHD (light encoder heavy decoder). The experiment results show that LEHD provides higher probing scores than AM and POMO, meaning that it embeddings are better in terms of (1) perception of Euclidean distance, (2) avoidance of myopia, and (3) perception of constraints.

**Strengths:**

S1. It is very interesting that the authors empirically show that the embeddings of LEHD capture better information than AM and POMO models.

S2. It seems that the authors properly design three probing tasks to examine the important abilities of high-performing NCO models.

**Weaknesses:**

W1. This paper can be considered as an analysis paper. In other words, this paper does not propose a novel idea for advancing NCO models.

W2. The probing tasks introduced in this paper mainly focuses on routing problems such as TSP (Traveling Salesman Problem), and CVRP (Capacitated Vehicle Routing Problem). However, there are many other CO problems like Knapsack and FFSP (Flexible Flow Shop Problem).

W3. Overall, this paper is well-written. However, it would be better to present more analysis results (in the Appendix) in the main sections, by slightly reducing the description of probing tasks.

**Questions:**

Q1. The datasets and models for probing tasks in this paper can help researchers and practitioners. Are the authors planning on making the datasets and models public?

---

> ### Comment · Reviewer_B7Vv · 2024-11-26
> **After the Author Response**
>
> Thank you for providing thoughtful responses to my comments. I maintain my initial score now. However, I am open to Area Chairs' opinions.

---

> > ### Author Response · Authors · 2024-11-26
> >
> > Thank you so much for your valuable feedback and support. Your insights are truly appreciated and help us improve and refine our work further.

---

### Official Review · Reviewer_wmJR · 2024-11-09

**Soundness:** 2
**Presentation:** 3
**Contribution:** 3
**Rating:** 5
**Confidence:** 4

**Summary:**

This paper introduces the novel use of probing techniques to interpret neural combinatorial optimization (NCO) models, particularly in the context of Traveling Salesman Problem (TSP) and Capacitated Vehicle Routing Problem (CVRP). The study aims to reveal the internal representations learned by three attention-based NCO models (AM, POMO, and LEHD). Through three distinct probing tasks, the authors assess each model's ability to perceive Euclidean distances, avoid myopic decisions, and handle capacity constraints. The findings highlight LEHD's superior performance in capturing decision-supporting information, which aligns with its structure featuring a light encoder and heavy decoder, as opposed to AM and POMO's heavy encoder-light decoder designs.

**Strengths:**

1)The application of probing techniques to NCO models is innovative, offering fresh insights into model interpretability.

2)The study is well-structured, with clear, contextualized findings that contribute to understanding NCO models.

3)The insights derived from probing tasks provide practical implications for NCO model design, particularly in enhancing decision-making and handling constraints.

**Weaknesses:**

1)Dataset limitations: The experiments are limited to relatively small instances, which may not fully capture the model's behavior on larger or more complex instances.

2)Comparison limited to LEHD and HELD architectures: The study compares only LEHD and HELD architectures, but since POMO and AM represent early work within the HELD category, they may lack sufficient representativeness for broader conclusions.

3)Difference in training methods: LEHD is trained using supervised learning, whereas POMO and AM are trained with reinforcement learning. The paper does not discuss whether the conclusions drawn are attributable to the model architectures or to the different training methods.

**Questions:**

1)In the ablation study section of the original LEHD paper, the authors discuss the impact of different training methods on model performance. I suggest adding ablation studies to further support your experimental results.

2)In Section 2, Probing Task 1, what are the specific implementation details of the proxy embedding?

3)In Table 1, Task 1 w/o ints., the R2 scores for AM and POMO are around 0.2. However, this lacks a reference point, making it difficult to gauge the model's learning degree for the Euclidean distance perception task. I suggest including randomly initialized, untrained versions of these models or models at various stages of training as a baseline for comparison. This would help demonstrate how the model's ability to perceive distance evolves over the training process.

4)In Table 1, some values are missing for Task 3. What is the reason for this?

5)In Lines 364-369, the description of the input for Task 3 is unclear. For the w/ints input, is it only [ hi ⊙ hj ]or [ hi , hj , hi⊙hj ]? In Table 5, you indicate it is the latter, but the text seems to suggest the former.

---

### Meta-Review · Area_Chair_gVat · 2024-12-17

**Metareview:**

This paper applies probing techniques to analyze NCO models, revealing their encoding of Euclidean distances and ability to avoid myopic decisions, while also exploring constraint understanding. Its strengths lie in the novel probing application and clear findings useful for model design. However, it has notable weaknesses, including small datasets, limited architecture comparisons, and unclear training method impacts. The probing tasks are somewhat simplistic and focused. Overall, the paper is rejected as the authors' revisions, while extensive, do not fully overcome these issues. Key concerns like experimental design flaws and lack of novelty compared to existing work remain, preventing it from meeting the acceptance threshold. In the reviewer discussion, concerns about dataset size, architecture diversity, and probing details were raised. The authors responded with new experiments and analyses. But in the final assessment, these efforts did not sufficiently address the fundamental problems. The work's contribution remains uncertain and its experimental basis is not strong enough, leading to the rejection decision.

**Additional Comments On Reviewer Discussion:**

During the rebuttal period, reviewers raised various points such as dataset limitations, the need for more diverse architectures in comparison, more detailed explanations of probing tasks and models, and expansion to other combinatorial optimization problems. The authors addressed these by conducting additional experiments (e.g., on new datasets and different models), providing more comprehensive result analyses across layers and epochs, and clarifying aspects like probing task implementation and the relationship between probing results and model performance. In the final decision, while the authors' efforts were noted, the outstanding concerns regarding the robustness and comprehensiveness of the study still weighed against acceptance as they impacted the overall validity and significance of the work.

---

### Decision · Program_Chairs · 2025-01-22

Reject